# Validation and Test of Measurement Invariance of the Adapted Health Consciousness Scale (HCS-G)

**DOI:** 10.3390/ijerph18116044

**Published:** 2021-06-04

**Authors:** Matthias Marsall, Gerrit Engelmann, Eva-Maria Skoda, Martin Teufel, Alexander Bäuerle

**Affiliations:** Clinic of Psychosomatic Medicine and Psychotherapy, LVR-University Hospital, University of Duisburg-Essen, 45147 Essen, Germany; gerrit.engelmann@lvr.de (G.E.); eva-maria.skoda@uni-due.de (E.-M.S.); martin.teufel@uni-due.de (M.T.); alexander.baeuerle@uni-due.de (A.B.)

**Keywords:** health consciousness, validation, health measurement, psychometric assessment, measurement invariance

## Abstract

The objective of this study was the translation and validation of a health consciousness scale in order to provide an economically and empirically confirmed measurement of health consciousness, which is associated with health-related behavior. We evaluated this translation on the basis of psychometric testing in a German convenience sample. A cross-sectional online survey (*n* = 470) was carried out using a translated version of the health consciousness scale, oriented on the basis of international guidelines. As previous studies have not consistently confirmed the factorial structure of the health consciousness scale, we conducted a Confirmatory Factor Analysis to verify its factorial structure. Furthermore, we cross-validated the questionnaire with other scales in order to verify convergent and discriminant validity. The results indicated a two-factor solution for the Health Consciousness Scale-German (HCS-G). The criterion validity was confirmed on the basis of a significantly positive correlation between the HCS-G and health literacy. Furthermore, strict measurement invariance was able to be verified, indicating that the HCS-G is an applicable measurement, regardless of gender. In practical research, this questionnaire can help to assess health consciousness and its influence on health-related constructs. Future studies should consider possible mediating variables between health consciousness and health outcomes.

## 1. Introduction

Health-related or preventive behaviors are an omnipresent topic in research. Meta-analyses as well as reviews have shown their influence on physical and mental health [1,2]. For example, reviews and different studies have found correlations between healthy diets and cardiovascular, metabolic (e.g., Diabetes mellitus), or oncological diseases [3,4,5]. Health-related and preventive behaviors such as physical activities are also recommended by international societies and studies, which have shown positive effects on disease prevention [6,7,8,9]. Furthermore, during the Covid-19 pandemic, preventive behavior seems key to protection from infection with the Covid-19 virus [10,11]. 

Different determinants of these health-related or preventive behaviors have been identified in studies [12,13,14,15]. These studies show that attitudes toward one’s own health or the perception of one’s health status are possible determinants of health-related or preventive behaviors. Constructs such as health consciousness [12], health orientation [16], health conception [17] and health locus of control [18] are psychological constructs that have been explored in health science that influence these behaviors. The construct of health consciousness, especially, focusses on one’s own perception of health status, and is described by Gould [12] as a self-awareness regarding one’s own health. Gould [12,19] distinguished between four different dimensions focused on an individual’s psychological inner state towards health-related awareness: health self-consciousness, health alertness, health self-monitoring and health involvement. Health consciousness and its interrelations with different health-associated outcomes has already been verified in various studies, such as with respect to different healthy activities [20], prevention of illness and promotion of long-term health [21], frequency of doctor visits [22] and less smoking/drinking, increased engagement with health information, and taking vitamin supplements [12]. Even though these outcomes were measured by self-reported questionnaires, relevant associations of health consciousness and health-associated outcomes were found. Furthermore, studies have shown that higher states of health consciousness are correlated with the likelihood of purchasing organic food and green furniture [23,24,25,26,27]. In this context, Shimoda and colleagues [28] additionally showed an association between health consciousness and pro-environmental behavior indicating that health consciousness is not only likely to promote one’s individual health but can also have positive environmental effects.

At this point it is important to distinguish the definition of health consciousness by Gould [12] from that of Jayanti and Burns [14]. Jayanti and Burns [14] define health consciousness as health concerns and their integration in daily behaviors or activities of people. This is a more action-oriented definition than the definition of Gould [12], who assumes health consciousness rather as a mindset influencing health-related and preventive behaviors. Furthermore, self-awareness is a key variable in the definition of health consciousness. 

Self-awareness is a self-directed attention to oneself and is associated with self-consciousness [12,29]. These constructs are the core of health consciousness. Thus, from our perspective, the definition of health consciousness and its operationalization according to Gould [12] offer the greatest opportunities to focus on self-awareness regarding one’s own health. The constructs self-awareness and self-consciousness are part of health consciousness and influence subsequent health-related actions and health behaviors [12,30]. Health consciousness, as a psychological determinant of health-related and preventive behavior, reflects a concrete attention towards one’s own health status and symbolizes a conscious treatment with one’s own health. Thus, it is a main variable influencing health-related and preventive behavior.

Gould’s scale [12,19] has already been used in many studies to measure health consciousness. However, the factorial structure has not been verified, and different factorial structures have been reported in several studies (e.g., [22,31,32]). According to our research, the factorial structure of Gould [12,19] has been confirmed in only one study with a first- and second-order factor structure [22]. However, looking at intercorrelations between second-order factors in Gould’s [12] factorial structure, it became evident that these factor loadings are too high to reflect independent factors. Here, a Confirmatory Factor Analysis (CFA) of the scale is needed to confirm the construct health consciousness in its subdimensions. Furthermore, the scale has not been used in German samples with a translation according to common scientific recommendations [33,34] and has not been validated in a German sample. 

It is important to ensure the construct validity of existing questionnaires. Therefore, the convergent as well as the discriminant validity of a questionnaire have to be verified. Another methodological approach focused on the verification of measurement invariance of the health consciousness scale is to confirm that the instruments measure the construct equally for both genders. To our knowledge, scientific evidence is still missing to assume measurement invariance of health consciousness between women and men. Nevertheless, statistical differences between different groups of people are not interpretable unless measurement invariance between these groups is confirmed [35]. Rather, a lack of measurement invariance of instruments may lead to misinterpretations of gender differences, as differences may be caused by the instrument itself [36]. Thus, Gould’s analysis of gender differences is not interpretable [12]. 

In fact, determinants of health-related and preventive behavior have been identified in different studies [12,13,14,15]. Thus, it is a key variable in the context of measuring one’s self-awareness towards one’s health status. However, a health consciousness scale in the German language does not exist and needs to be validated to focus on its impact on health-related and preventive behavior in the German population. Furthermore, different studies (e.g., [22,31,32]) using the health consciousness scale have shown ambiguous factorial structures, which have seldom been validated, while only rarely representing the assumed factorial structure of Gould [12]. 

In summary, we are appealing to different aims by conducting this validation study of a psychometric instrument measuring health consciousness. Due to the practical and scientific demands and the described limitations, we focused on four objectives. First, we wanted to develop a validated and a translated version of the health consciousness scale in German language according to common scientific recommendations [33,34] to ensure content validity. Second, we aimed to verify a valid factorial structure of this new instrument. For this purpose, we performed Confirmatory Factor Analysis (CFA) to examine the factorial structure of the translated instrument. Additionally, to confirm construct validity, we scrutinized convergent and discriminant validity. To verify convergent validity, we assumed a positive correlation between health consciousness and personality trait conscientiousness, because both constructs describe a person’s tendency to be aware of and responsive to one’s surroundings in a thoughtful way. We assumed that personality trait openness would be associated with health consciousness, because both constructs symbolize a person’s interest in searching for and discovering changes and improvements in one’s life with respect to health status and life situations (e.g., [37]). Moreover, as health consciousness focuses on a mindful handling of one’s body, we expected a negative interrelation with impulsivity. To verify discriminant validity, we looked at interrelations with different personality traits such as extraversion and neuroticism, assuming no significant correlations, as there were no content-related overlaps with health consciousness. Third, we wanted to ensure the criterion validity of the adapted instrument. For this purpose, we asked participants about their health literacy and the frequency with which they had used medical or therapeutic help within the previous twelve months. As in the original study of Gould [12], there were, unexpectedly, no significant interrelations found between health consciousness and physical or mental health, we captured these constructs as well. Fourth, we wanted to perform an analysis of measurement invariance of the Health Consciousness Scale—German (HCS-G) regarding gender. To our knowledge, there is no study examining measurement invariance of health consciousness between women and men. Nevertheless, a lot of studies have shown differences between women and men regarding their health and preventive behavior, such as their use of health services [38], their attitude towards seeking psychological help [39], and health status [40]. However, interpretation of gender differences should be made with caution when the measurement invariance of the applied instruments has not been confirmed. Therefore, it is essential to ensure measurement invariance between women and men, as increasingly frequently examined in recent studies (e.g., [41,42,43]).

## 2. Methods

### 2.1. Adaption of the HCS-G and Study Design 

We conducted the translation following the guidelines for translation of academic literature to ensure content validity [33,34]. The translation was conducted in four steps. In a first step, the items were translated into German by two authors, and these translations were merged into a first translation proposal. In the second step, a systematic expert panel consisting of the two translators and two psychologists discussed the merged items. The two psychologists were experts within the context of healthcare. In a third step, the resulting items in the German language were translated back into English by an English native speaker, confirming that their meaning was consistent with the original items. In a fourth step, cognitive interviews were conducted to ensure easy understanding, and inoffensive and non-discriminatory phrasing of the items. The resulting final version of the translated questionnaire consisted of nine items. The original items, as well as the translated items, are displayed in Appendix A. We used a 5-point Likert scale from 1 = “strongly disagree” to 5 = “strongly agree”. A sample item is “I’m very involved with my health”. We clustered the items according to the assumed factorial structure of Gould (1990): health self-consciousness, health alertness, health self-monitoring and health involvement [12]. 

Our cross-sectional online survey was conducted between October 2020 and November 2020 in a German-speaking convenience sample. Participants were recruited via online social networks like Xing, LinkedIn, and Facebook. All data were collected anonymously. We considered completed data sets only. We conducted the study according to the guidelines of the Declaration of Helsinki and the Ethics Committee of the Medical Faculty of the University of Duisburg–Essen approved our survey (approval number 20-9592-BO).

### 2.2. Participants

524 participants completed our questionnaire which represents a completion rate of 32%. Participants who took less than 5:34 min (5% percentile) or more than 25:45 min (95% percentile) to complete the survey were excluded from the analysis. We decided not only to exclude extreme fast responders but also extreme slow responders as we wanted to ensure that our analysis was based on an average sample where possible biases in response behavior were minimized. Furthermore, we excluded one participant for being under 18 years of age. One person indicated their gender as “diverse”, which we excluded to perform the analysis of measurement invariance of gender. The resulting sample consisted of *n* = 470 participants, which is in accordance with the sample size recommendation for test validation [44,45].

### 2.3. Study Variables

The following measurements were captured to cover the study objectives. Prior to the statistical analyses, inverted items were recoded. 

#### 2.3.1. Health Literacy

The Health Literacy Questionnaire (HLS-EU-Q16) measures health literacy with 16 items [46]. A sample item is “How easy/difficult is it to find information on treatments of illnesses that concern you?”. Health literacy was measured on a two-point scale (easy/hard), resulting in a sum-score of health literacy between 0 and 16.

#### 2.3.2. Impulsivity

The Impulsive Behavior-8 Scale (I-8) measured impulsivity using eight items (e.g., “Sometimes I spontaneously do things that I should not have done” [47].

#### 2.3.3. Personality Traits

The BFI-10 [48] measures personality traits (extraversion, neuroticism, openness, conscientiousness, agreeableness), which are each assessed by two items. A sample item for neuroticism is “I get nervous and insecure easily”. The reliabilities of conscientiousness and agreeableness were low, as the items capturing these constructs represent different contents of these traits. One item of conscientiousness measures if one tends to laziness, while the other measures if one completes tasks thoroughly. As we judged both aspects of conscientiousness to be highly relevant in the context of health consciousness, we decided to use the single items in further analyses. As agreeableness was not regarding as contributing to the validation of health consciousness, it was excluded from further analyses.

#### 2.3.4. Further Constructs

Physical health and mental health were measured using self-developed single items on an 11-point Likert scale (0 = “very bad health” to 10 = “very good health”). Furthermore, we asked participants about their frequency of use of medical or therapeutic help within the previous twelve months from 1 = “never” to 6 = “every week”.

Additionally, sociodemographic variables (age, gender, marital status, educational level, financial situation, and residential area) were assessed.

### 2.4. Statistical Analysis

All statistical analyses were performed with R, RStudio and additional packages [49,50,51,52,53,54,55]. We first examined sample characteristics and item statistics of the HCS-G. To investigate the factorial structure of the health consciousness items, we conducted several Confirmatory Factor Analyses. We considered the recommendations of Hu and Bentler [56] for model evaluation. We examined Comparative Fit Index (CFI), Tucker Lewis index (TLI), Root Mean Square Error of Approximation (RMSEA), and Standardized Root Mean Square Residual (SRMR), as well as the factor loadings of items, to determine which model fits best to the data. As we collected survey data, we assumed data to be ordinal level rather than interval level. The use of Weighted Least Square Mean and Variance Adjusted estimator (WLSMV) is recommended for ordinal data rather than the Maximum Likelihood Robust estimator (MLR) in terms of the robustness of estimations of factor loadings and model fit indices [57]. Therefore, we conducted the CFA using the WLSMV estimator. However, WLSMV tends to overestimate inter-factor correlations, which will be taken into account in the discussion section [57,58]. Furthermore, we calculated Cronbach´s alpha (CA) with 95% confidence interval boundaries (CI) to report the reliability of the HCS-G, as well as the scales for construct and criterion validation. 

To scrutinize the convergent, discriminant, and criterion validity of health consciousness, we performed two-tailed Pearson correlations and considered correlations to be significant at *p* < 0.05. 

We examined measurement invariance to test whether the measurement of health consciousness was equally appropriate for both genders. The test of measurement invariance follows four sequential steps. Each step assumes a more constrained model. In step one, the number of factors and the factor-to-indicator relationship were considered to be equal between genders, which describes configural invariance. In step two, we scrutinized metric invariance by additionally keeping factor loadings equal between genders. Moreover, in step 3 (scalar invariance), intercepts are kept equal. Finally, the most constrained model 4 (conservative invariance) assumes the equality of residual variances. A more constrained model can be accepted in cases where the differences between it and the less constrained model are low [59,60]. The WLSMV estimator is also used for measurement invariance in order to compare the different models. As the interpretation of results of measurement invariance using the WLSMV estimator is limited, especially regarding changes in Chi^2^, we focus on changes in CFI and the other fit indices [61,62,63].

## 3. Results

### 3.1. Study Sample

After the described case exclusion, average completion time of our survey was 11:32 min (SD = 4:24 min, Median = 10:38 min). The mean age of our participants was 37.2 (SD = 13.4, Median = 33, Minimum = 18, Maximum = 82). Table 1 shows detailed sociodemographic characteristics of the study sample.

### 3.2. Item Statistics

Table 2 depicts the item statistics of the nine items of the preliminary HCS-G. The last characters indicate the subscales as proposed in the original study (C = self-consciousness; I = involvement; A = alertness; M = self-monitoring).

All item distributions are slightly negatively skewed. Items hcon3C (health self-consciousness), hcon5A and hcon6A (both health alertness) showed high values of kurtosis. Response distribution and SD indicated that these items were poor in generating variance between participants.

### 3.3. Confirmatory Factor Analyses

In our first model, all nine items loaded on a single factor. Model 2 considered the proposed subscales of the original study from Gould, 1990 [12]. Due to poor item statistics of three items (hcon3C, hcon5A and hcon6A) and their low factor loadings, we conducted a third model as a single factor model with remaining items. We verified a fourth model as a three-factor model with intercorrelated factors. Since in this model the subscales of health self-consciousness and health involvement were highly correlated (*r* = 0.93, *p* < 0.001), we carried out a fifth CFA as a two-factor model with items of health self-consciousness and health involvement considered as one factor. A second-level factor of health consciousness led to convergence problems just as Gould reported in his study [12]. Therefore, we assumed health consciousness to consist of two intercorrelated constructs which measure different aspects of health consciousness. Table 3 shows the results of performed CFA.

In the three-factor (short) model, reliabilities for health self-consciousness, health involvement, and health self-monitoring were 0.74 (CI = 0.69–0.79), 0.80 (CI = 0.76–0.83), and 0.84 (CI = 0.81–0.87), respectively. Within the two-factor (short) model, items of health self-consciousness and health involvement were considered to load on a single factor, which reached a reliability of 0.85 (CI = 0.83–0.88). Considering these results, the two-factor (short) model was assumed to best represent the construct of health consciousness. In fact, the three-factor (short) model had partly better fit indices than the two-factor (short) model. Nevertheless, the correlation of 0.93 between self-consciousness and involvement strongly indicated that those two subscales were measuring the same content. Moreover, the two-factor (short) model was a less constrained model, representing a more applicable and parsimonious model.

Figure 1 shows the final two-factor (short) model with factor loadings as well as inter-factor correlation.

### 3.4. Validation Analyses

As described, our CFA process found the two-factor (short) model with intercorrelated factors to best fit the data. Therefore, we conducted the following analyses, considering the two identified factors self-consciousness (M = 3.81, SD = 0.77) and self-monitoring (M = 3.40, SD = 0.89) as independent variables.

#### 3.4.1. Health Consciousness and Sociodemographic Variables

To test whether health consciousness differs between genders, two-sided t-tests were conducted. There was no significant difference between men and women in self-consciousness (*t* = 1.37, *p* = 0.17) or self-monitoring (*t* = 0.30, *p* = 0.77). Further interrelations between health consciousness and sociodemographic variables are shown in Table 4.

The results show that there are no significant correlations between self-consciousness or self-monitoring and sociodemographic variables, except for educational level, which correlated significantly negative with self-monitoring (*r* = −0.13, *p* = 0.006).

#### 3.4.2. Validation of Health Consciousness

Table 5 depicts Pearson correlations of health consciousness scales and conscientiousness (single items), openness (M = 3.61, SD = 0.99, CA = 0.62, CI = 0.54–0.68), impulsivity (M = 2.78, SD = 0.59, CA = 0.72, CI = 0.68–0.76), extraversion (M = 3.30, SD = 1.04, CA = 0.79, CI = 0.76–0.83), and neuroticism (M = 3.08, SD = 0.97, CA = 0.66, CI = 0.60–0.72). 

Looking at convergent validity, both HCS-G scales correlated significantly positively with openness and significantly negatively with impulsivity. Furthermore, both items of conscientiousness were significantly interrelated with self-consciousness, but not with self-monitoring. Considering discriminant validity, correlations of both HCS-G scales with personality traits of extraversion and neuroticism were not significant.

Regarding criterion validity, correlations between the health consciousness scales and the health-related construct of health literacy (M = 12.63, SD = 2.99, CA = 0.79, CI = 0.76–0.81), as well as the frequency of use of medical or therapeutic help (M = 3.80, SD = 1.50) are shown in Table 6.

As in the original study by Gould [12], we additionally examined interrelations of health consciousness scales and physical health (M = 7.37, SD = 1.58), as well as mental health (M = 7.27, SD = 1.90). Neither self-consciousness (*r* = −0.02, *p* = 0.74 and *r* = 0.05, *p* = 0.32) nor self-monitoring (*r* = 0.00, *p* = 0.97 and *r* = 0.01, *p* = 0.83) correlated significantly with physical or mental health.

### 3.5. Measurement Invariance

The tests of measurement invariance regarding the gender of participants were conducted on the final two-factor model. Table 7 shows the results of these analyses.

At all levels of measurement invariance, only small changes in CFI and the other fit indices could be observed. Improvement of RMSEA can be explained by the sensitivity of this fit index to small degrees of freedom [64].

## 4. Discussion

The most essential strengths of our study are due to the high methodological and psychometrical standards applied to the validation and adaptation of the HCS-G. In this regard, this study confirmed its content, construct, and criterion validity. Another main strength of this study is the confirmation of measurement invariance with respect to gender. To our knowledge, no other study has examined the measurement invariance of the health consciousness construct or revealed methodological insufficiencies.

Our first aim was to develop a valid version of the health consciousness scale in the German language and to confirm the content validity of our new instrument, which we achieved by strongly following the scientific recommendations implied by the development of the HCS-G. To achieve our second aim, we validated our new instrument by performing several Confirmatory Factor Analyses to examine the factorial structure. Within the correlational analyses, we verified convergent and discriminant validity with respect to construct validity. We followed a stepwise CFA approach in order to acquire the best-fitting and most parsimonious model. Therefore, we excluded single items, resulting in a six-item model that represents health consciousness on two subscales. To our knowledge, this is the first study to critically examine the assumed factorial structure according to Gould [12]. A two-factorial structure best represented the construct of health consciousness, while also solving the convergence problems of Gould’s original instrument [12] and confirming a valid factorial structure which has been neglected in other studies so far [21,32,65,66,67]. Our final model achieved good fit indices and factor loadings of the items representing the factors self-consciousness and self-monitoring. A slight to high value of RMSEA can be explained by its sensitivity to small degrees of freedom, which was true for the tested model [64]. Both factors were significantly positively interrelated, but represented different aspects of health consciousness, as confirmed by our correlational analyses. Self-consciousness can be understood as the awareness of one’s health in the long term, whereas self-monitoring focuses more on daily routines of self-observation.

As expected, the HCS-G scales were significantly positively correlated with consciousness and openness, as well as being significantly negatively correlated with impulsivity, indicating convergent validity. This indicates that a greater score in the HCS-G scales is associated with greater conscientiousness and openness, and lower impulsivity. Furthermore, the HCS-G scales did not correlate with constructs that were different from health consciousness (extraversion and neuroticism), confirming discriminant validity. Surprisingly, HCS-G (sm) was significantly negatively correlated with educational level. This result replicates the findings of Gould [12], who suggested that people with lower educational levels may spend more time on their health, resulting in a higher health consciousness. Nevertheless, a valid explanation of this relationship is still lacking. One could also think that conceptions regarding one’s health and health consciousness might carry different meanings in different educational levels, resulting in different relational patterns. However, as our sample mainly consisted of higher-educated people, it was not possible to administer an in-depth analysis within a subsample of lower-educated people. Hence, future studies should examine this relationship in lower-educated subsamples or from a causal viewpoint.

Regarding our third aim, the criterion validity of the HCS-G was confirmed by verifying a positive correlation between both scales of health consciousness and health literacy. This indicates that a greater score in the HCS-G scales is associated with greater health literacy, which is in agreement with the results of Gould [12]. It is possible that people with greater health consciousness examine their health more precisely and look for health information from different sources in order to assess their own state of health. Furthermore, as health literacy can be understood as a competence in dealing with one’s health issues, which is associated with many health-related outcomes (e.g., [68,69,70]), health consciousness is a highly relevant construct in the understanding of a holistic concept of people’s health. However, only HCS-G (sc) was significantly correlated with the use of medical or therapeutic assistance, indicating that highly self-conscious people use medical or therapeutic help more often. HCS-G (sm) was not significantly correlated with the frequency of medical or therapeutic help, which could be explained by the fact that self-monitoring items tend to focus on daily routines, while self-consciousness items tend to be oriented towards long-term aspects. Unexpectedly, health consciousness was not correlated with possible outcome variables of physical and mental health, which is in agreement with the results of Gould [12]. We assumed health consciousness to be related to higher physical and mental health statuses. To explain these findings, one could think that people with lower subjective health statuses may be forced to become more concerned with their health, leading to higher scores in health consciousness. Furthermore, several studies have shown that conceptualizations, attitudes, and intentions are not causally linked to actual health behavior [71,72,73]. In consequence, the relationship between health consciousness and health-related outcomes may be mediated by other variables. Future studies should investigate possible mediating variables between health consciousness and health outcomes.

In the case of the Covid-19 pandemic, health consciousness could be a relevant psychological construct for preventing further Covid-19 outbreaks. In our study, health consciousness and health literacy are associated constructs. Thus, one could assume that health-conscious people will be more likely to be self-aware and vigilant with respect to diseases and their influences on the body. Furthermore, health-conscious people will be more likely to understand Covid-19 interventions and to comply with the regulations, resulting in a reduced spread of Covid-19. The compliance with the regulations [74] is a relevant indicator for the prevention of an infection. Thus, health-conscious people could be more likely to successfully avoid Covid-19 infection, e.g., because of a higher adherence to safety behaviors [75]. Future studies should take the associations between health consciousness and infection prevention, especially infection with Covid-19, into account precisely because health consciousness could be a relevant mediator in the association between prevention knowledge and prevention behavior in Covid-19 infection [76].

Our fourth aim contributed an examination of the measurement invariance of health consciousness between women and men. The measurement invariance of the HCS-G with respect to gender was confirmed, verifying that the HCS-G is a reliable instrument for measuring health consciousness and interpreting gender differences. Measurement invariance is an essential requirement when measurements are used to interpret differences between groups. Through the confirmation of the measurement invariance of the HCS-G with respect to gender, the interpretation of differences in gender can be regarded as valid, i.e., the differences found are not a result of the measurement method itself. Therefore, our instrument provides researchers as well as practitioners the opportunity to capture and interpret gender differences with respect to health consciousness, as well as to develop and evaluate interventions in consideration of gender. In conclusion, the HCS-G is an ideal instrument for practical applications, as the interpretation of the results is carried out independently of gender. As health consciousness is a relevant construct with respect to both maintaining health and influencing health-related behavior, the HCS-G can be used during the diagnostic process as one of the resources for patients. By considering two aspects of health consciousness, the HCS-G can also be used to develop and evaluate psychological training with the aim of improving health consciousness.

Nevertheless, limitations have to be considered in this study. One limitation is related to the cross-sectional study design. Thus, correlations are not causally interpretable, but show relationships between the different measured constructs. A second limitation with respect to our study design is related to the method of data collection being via an online survey, which could lead to a selection bias. It is likely that only participants who were familiar with the use of the Internet took part in our study. A third limitation is related to our distribution of educational levels: a high proportion of people that took part in our study held a university degree, limiting the representativeness with respect to educational level. Even though it was our goal to collect data on a convenience sample representing the German population, the study sample did not reflect a balanced gender distribution (71% female participants).

## 5. Conclusions

Health consciousness is a highly relevant psychological construct in the context of several health-related outcomes and in examining people’s health. We conducted our study with a focus on the highest methodological and sychometrical standards in order to provide a valuable opportunity to measure health consciousness, especially when it comes to the interpretation of gender differences. Until now, a deeper understanding of the factorial structure, as well as the validity, of the measurement of health consciousness has been missing. The HCS-G fills this important gap and enhances the field of application by confirming the measurement invariance. Furthermore, the HCS-G is the first instrument to measure this highly relevant construct in the German language.

## Figures and Tables

**Figure 1 ijerph-18-06044-f001:**
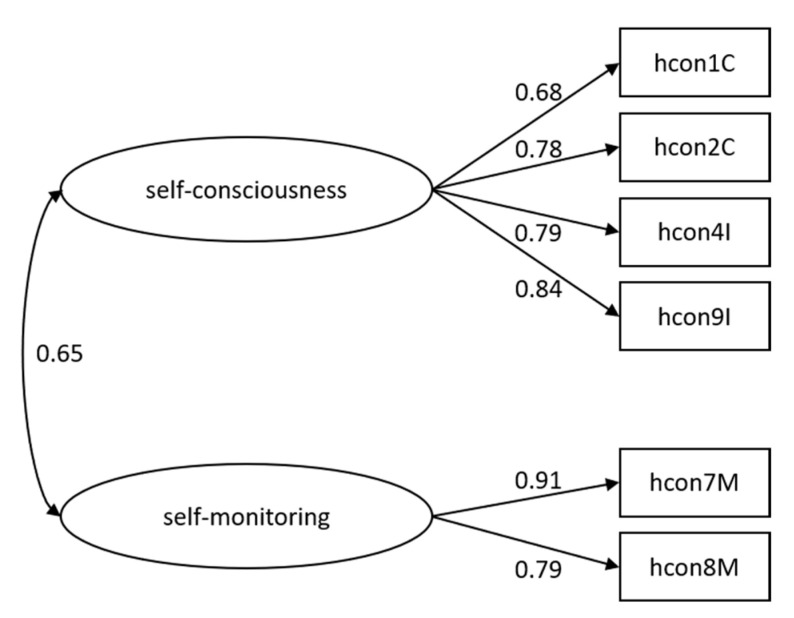
Two-factor (short) model of health consciousness. Square boxes represent items. Circles represent factors. Unidirectional arrows indicate factor loadings. Bidirectional arrow indicates inter-factor correlation.

**Table 1 ijerph-18-06044-t001:** Detailed sociodemographic characteristics of the study sample.

		*n*	%
gender	female	332	71%
	male	138	29%
marital status	married or in partnership	344	73%
	single	115	24%
	other	11	2%
educational level	university degree	273	58%
	completed vocational training	91	19%
	maturity for university	77	16%
	secondary school certificate	29	6%
subjective financial situation	very good or good	300	64%
	average	114	24%
	bad or very bad	56	12%
residential area	big city above 100,000 residents	244	52%
	city above 20,000 residents	88	19%
	town or village up to 20,000 residents	138	29%

**Table 2 ijerph-18-06044-t002:** Item statistics for all items of the health consciousness scale.

Item	Mean	SD	Median	Skew	Kurtosis	Response Distribution				
						1	2	3	4	5
hcon1C	3.88	0.89	4	−0.93	0.72	1%	9%	12%	56%	22%
hcon2C	3.89	0.88	4	−0.89	0.65	1%	9%	12%	56%	22%
hcon3C	4.07	0.8	4	−1.22	2.25	1%	6%	6%	60%	27%
hcon4I	3.71	0.97	4	−0.74	−0.02	2%	13%	15%	52%	18%
hcon5A	4.1	0.72	4	−1.06	2.42	0%	4%	7%	63%	26%
hcon6A	4.19	0.67	4	−0.84	1.71	0%	3%	6%	61%	30%
hcon7M	3.23	1.01	3	−0.23	−0.79	3%	24%	26%	39%	8%
hcon8M	3.56	0.91	4	−0.61	−0.14	1%	14%	22%	52%	11%
hcon9I	3.75	0.92	4	−0.62	−0.25	0%	13%	16%	51%	19%

**Table 3 ijerph-18-06044-t003:** Results of Confirmatory Factor Analysis examining the factorial structure of the Health Consciousness Scale—German (HCS-G).

Model	Chi^2^	df	CFI	TLI	RMSEA	SRMR
(1) single factor	229.990	27	0.763	0.684	0.127	0.077
(2) four-factor	114.590	21	0.891	0.813	0.097	0.056
(3) single factor (short)	193.054	9	0.695	0.491	0.209	0.087
(4) three-factor (short)	34.200	6	0.953	0.883	0.100	0.035
(5) two-factor (short)	37.854	8	0.950	0.907	0.089	0.038

df = degree of freedom. CFI = Comparative Fit Index. TLI = Tucker Lewis index. RMSEA = Root Mean Square Error of Approximation. SRMR = Standardized Root Mean Square Residual. short = models considering six items.

**Table 4 ijerph-18-06044-t004:** Person correlations of health consciousness scales and sociodemographic variables.

	HCS-G (sc)	HCS-G (sm)	Age	Marital Status	Educational Level	Financial Situation
HCS-G (sm)	0.55 ***					
age	0.07	0.02				
marital status	0.04	−0.04	0.02			
educational level	−0.06	−0.13 **	−0.04	−0.08		
financial situation	−0.05	−0.08	0.12 **	−0.12 **	0.22 ***	
residential area	0.01	0.03	0.13 **	0.03	−0.21 ***	0.00

HCS-G (sc) = Health Consciousness Scale—self-consciousness. HCS-G (sm) = Health Consciousness Scale—self-monitoring. ** *p* < 0.01, *** *p* < 0.001.

**Table 5 ijerph-18-06044-t005:** Pearson correlations of health consciousness scales and convergent and discriminant scales.

	HCS-G (sc)	HCS-G (sm)	consc.1	consc.2	Openness	Impulsivity	Extraversion
HCS-G (sm)	0.55 ***						
consc.1	0.15 **	0.08					
consc.2	0.13 **	0.08	0.24 ***				
openness	0.12 *	0.10 *	0.00	0.08			
impulsivity	−0.14 **	−0.11 *	−0.27 ***	−0.46 ***	0.05		
extraversion	0.03	0.03	0.07	−0.17 ***	0.08	0.26 ***	
neuroticism	0.01	−0.05	−0.08	0.11 *	−0.07	−0.08	−0.30 ***

HCS-G (sc) = Health Consciousness Scale—self-consciousness. HCS-G (sm) = Health Consciousness Scale—self-monitoring. consc.1 = conscientiousness item 1 (recoded): “I am comfortable, prone to laziness” (M = 3.18, SD = 1.10). consc.2 = conscientiousness item 2: “I complete tasks thoroughly” (M = 4.01, SD = 0.80). * *p* ≤ 0.05, ** *p* < 0.01, *** *p* < 0.001.

**Table 6 ijerph-18-06044-t006:** Pearson correlations of health consciousness and criterion validity scales.

	HCS-G (sc)	HCS-G (sm)	Health Literacy
HCS-G (sm)	0.55 ***		
health literacy	0.09 *	0.15 **	
frq. med. help	0.09 *	−0.01	0.06

HCS-G (sc) = Health Consciousness Scale—self-consciousness. HCS-G (sm) = Health Consciousness Scale—self-monitoring. Frq. Med. Help = frequency of medical or therapeutic help. * *p* ≤ 0.05, ** *p* < 0.01, *** *p* < 0.001.

**Table 7 ijerph-18-06044-t007:** Measurement invariance of the final HCS-G model regarding gender.

Level of Invariance	Chi^2^	df	CFI	TLI	RMSEA	SRMR	Δ CFI
configural	45.123	16	0.952	0.910	0.088	0.035	−0.002
metric	36.226	20	0.973	0.960	0.059	0.038	0.021
scalar	42.600	24	0.969	0.962	0.058	0.041	−0.004
conservative	47.259	30	0.972	0.972	0.050	0.045	0.003

df = degree of freedom. CFI = Comparative Fit Index. TLI = Tucker Lewis index. RMSEA = Root Mean Square Error of Approximation. SRMR = Standardized Root Mean Square Residual. Δ CFI = change in CFI compared to less constrained model.

## Data Availability

The data presented in this study are openly available in FigShare: Dataset: https://doi.org/10.6084/m9.figshare.14502267 (accessed on 30 January 2021). R-Syntax: https://doi.org/10.6084/m9.figshare.14502276 (accessed on 30 January 2021).

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
