# Peer review of "Validation and Test of Measurement Invariance of the Adapted Health Consciousness Scale (HCS-G)"

_ijerph, 2021, doi:10.3390/ijerph18116044_

Round 1

Reviewer 1 Report

In the present research, Matthias Marsall and colleagues performed an adaptation to German language and validation of the Health Consciousness Scale with the aim of providing an economic and empirically confirmed measurement of health consciousness. The study concluded that the scale is helpful in assessing health consciousness and its influence to health-related constructs.

By doing so, the authors valuably contribute to an important topic, providing a promising instrument to measure health consciousness and highlighting its fundamental role. The sample size is appreciable.

The added value of this work lies in the fact that the authors conducted a confirmatory analysis to verify the factorial structure of the scale, considering that previous research failed to consistently confirm it.

The work appears to have been conducted straightforwardly.

Introduction: it provides a clear background, but I advise the authors to shorten it a little bit.

Methods: adaptation of the scale and study design seems appropriate and are well reported. Conversely, the “Participants” subsection should only describe inclusion and exclusion criteria, while sample description should be moved to Results (3.1 Study sample) following the STROBE statement for observational studies (https://www.strobe-statement.org/index.php?id=available-checklists). The same applies for subsection 2.3 and all the relevant paragraphs; scores and reliability coefficients should be moved to the Results section, as these represent findings of the research and not methods.

Results: see comments about Methods. Otherwise, they are clear and well presented. Confidence intervals for Cronbach’s alpha should be displayed to provide the reader with more precise information.

Discussion: no major concerns. It is well-balanced between discussing the results of the study and making wider-ranging considerations. It has a pleasant logic flow. Reference to relevance of health consciousness during COVID-19 pandemic is timely.

Comprehensively, English quality is very good.

I have no major reasons for concern about how the research was conducted, but the abovementioned issues should be fixed in order to further improve the quality of the manuscript.

Reviewer 2 Report

General Comments

Thank you for the opportunity to review your paper. The topic of the paper is interesting and if appropriate revisions are made could fit the scope of the journal which is to focus “... on the publication of scientific and technical information on the impacts of natural phenomena and anthropogenic factors on the quality of our environment, the interrelationships between environmental health and the quality of life, as well as the socio-cultural, political, economic, and legal considerations related to environmental stewardship, environmental medicine, and public health”.

Specific Comments

  • My main concern on your study is that I could not found the link with the scope of the journal, I would suggest modifying your introduction and discussion sections appropriately.
  • L 13: Please consider changing capital “N” (population) with small “n” indicating sample size throughout the manuscript.
  • L 181: Please consider providing more information regarding the sample size selection. Appropriate power analysis is needed.
  • L182: Although it is clear why you excluded the fastest 5% of your responders, it is unclear why you chose to exclude the slowest ones as well. Probably the people who needed more than 25:45 minutes to complete the survey were older individuals or people with limited knowledge on computers. Please consider providing more information on this.
  • L 186: Same as the first comment.
  • L 187-188: This average value is much closer to the lowest 5% of the responders, indicating very large variance. I would suggest presenting Median as well.
  • L 246-247: Please consider changing as follows: “… to be significant at p < 0.05.”.
  • 261-397: Results sections should be dedicated on your findings. You presented all the statistical analyses conducted in the section “2.4 Statistical Analysis”, therefore please consider omitting mentioning the statistical analyses performed in the Results section.
  • Notes in your Tables could be clearer. For instance, I feel that it would be clearer to the reader if you directly explain that “df” indicates “degree of freedom” compared to the alphabetic symbols currently used.

Reviewer 3 Report

Thank you for submitting your paper for review and for conducting this interesting study. The objective of this study was the translation and validation of the health consciousness scale to provide an economic and empirically confirmed measurement of health consciousness, which is associated with health-related behavior. Health consciousness is a highly relevant psychological construct in the context of 506 several health-related outcomes and in examining people’s health. There is interesting data in this study, the authors have conducted the study with focus on highest methodological and psychometrical standards to provide a valuable opportunity in measuring health consciousness.

The manuscript is well-written and the English language is accurate both linguistically and regarding scientific purposes. I would like to highlight an additional point that will need to be addressed:

References. Please, check the final references, some journals' titles have not been abbreviated.

Round 2

Reviewer 2 Report

The authors have adequately answered all of my comments. Thank you very much for giving me the opportunity to review your work.